 eLIFE

# Magnetosensitive neurons mediate geomagnetic orientation in *Caenorhabditis elegans*

**Andrés Vidal-Gadea[1†], Kristi Ward[1], Celia Beron[1], Navid Ghorashian[2], Sertan Gokce[3], Joshua Russell[1], Nicholas Truong[1], Adhishri Parikh[1], Otilia Gadea[1], Adela Ben-Yakar[2], Jonathan Pierce-Shimomura[1]\***

[1]Department of Neuroscience; Center for Brain, Behavior and Evolution; Center for Learning and Memory; Waggoner Center for Alcohol and Addiction Research; Institute of Cell and Molecular Biology, University of Texas at Austin, Austin, United States; [2]Department of Mechanical Engineering, University of Texas at Austin, Austin, United States; [3]Department of Electrical Engineering, University of Texas at Austin, Austin, United States

**Abstract** Many organisms spanning from bacteria to mammals orient to the earth's magnetic field. For a few animals, central neurons responsive to earth-strength magnetic fields have been identified; however, magnetosensory neurons have yet to be identified in any animal. We show that the nematode *Caenorhabditis elegans* orients to the earth's magnetic field during vertical burrowing migrations. Well-fed worms migrated up, while starved worms migrated down. Populations isolated from around the world, migrated at angles to the magnetic vector that would optimize vertical translation in their native soil, with northern- and southern-hemisphere worms displaying opposite migratory preferences. Magnetic orientation and vertical migrations required the TAX-4 cyclic nucleotide-gated ion channel in the AFD sensory neuron pair. Calcium imaging showed that these neurons respond to magnetic fields even without synaptic input. *C. elegans* may have adapted magnetic orientation to simplify their vertical burrowing migration by reducing the orientation task from three dimensions to one.

*For correspondence: jonps@austin.utexas.edu

Present address: †School of Biological Sciences, Illinois State University, Normal, United States

Competing interests: The authors declare that no competing interests exist.

## Introduction

Many organisms such as birds, butterflies and turtles use the magnetic field of the earth (geomagnetic field) to navigate across the globe (*Johnsen and Lohmann, 2005*; *Guerra et al., 2014*). Many animals migrate horizontally by preferentially using either the horizontal (e.g., salmon, *Quinn et al., 1981*), or the vertical component of the earth's field (e.g., turtles, *Light et al., 1993*). By contrast, magnetotactic bacteria use the geomagnetic field to migrate roughly vertically by following the magnetic dip line (*Blakemore, 1975*). Across hemispheres, magnetotactic bacteria reverse their polarity-seeking preference, thus conserving the adaptiveness of the response (*Blakemore et al., 1980*).

Although much is known about magnetosensation in bacteria, the cellular and molecular basis for magnetosensation in animals is gaining in understanding. Recent progress has been made identifying central neurons that respond to magnetic fields (e.g., *Wu and Dickman, 2012*). Moreover, advancements have also been made identifying candidate magnetosensory transduction mechanisms (e.g., *Gegear et al., 2010*; *Lauwers et al., 2013*). Despite this progress, no magnetosensory neurons have been identified in any animal (*Edelman et al., 2015*). Understanding how animals detect and use magnetic fields will allow us to better predict the behavior of magnetosensitive organisms, and will aid the study of how natural and artificial magnetic fields affect living systems (*Engels et al., 2014*).

**eLife digest** The Earth has a magnetic field that protects the planet from the harmful effects of cosmic rays, which is generated by the movement of the layer of molten metal that surrounds the planet's solid inner core. The orientation of the magnetic field relative to the Earth's surface varies around the globe, and is like the pattern adopted by iron filings around a bar magnet.

Many organisms, from bacteria to birds such as the Arctic Tern and mammals such as wolves, are able to exploit this variation in the Earth's magnetic field to help them navigate. In bacteria, this ability has been linked to the possession of tiny magnetic particles that align with the Earth's magnetic field lines. However, it is not clear how animals are able use the magnetic field to navigate.

Vidal-Gadea et al. have now obtained clues to this process from a surprising source, namely a soil-dwelling nematode worm called *Caenorhabiditis elegans*. Worms that were placed inside a gelatin-filled cylinder preferred to burrow downward, but exposure to an artificial magnetic field that in effect reversed the Earth's magnetic field caused them to burrow upward. By contrast, worms that had been well-fed burrowed in the opposite direction.

When worms were instead allowed to crawl across a flat surface with a parallel magnetic field, a map of the Earth's magnetic field revealed that the worms, which were originally from England but which were tested in the US, moved at an angle that would correspond—in England—to burrowing downwards when they were hungry and upwards when were not. Consistent with this, worms from Australia crawled in the opposite direction to English worms, setting off at an angle that would also be equivalent—in Australia—to downwards when hungry and upwards when full. Identical results were found for Hawaiian worms.

Worms from countries on the Equator were less sensitive to magnetic fields than their northern and southern counterparts; which suggests that the response is genetically encoded. And indeed, a survey of mutant worms with defects in different neurons uncovered a pair of sensory neurons called 'AFD neurons' that serve the purpose of a compass, and allow the animals to navigate using geomagnetism.

It remains unclear exactly how AFD neurons detect the Earth's magnetic field, and it is also not clear why hungry worms should burrow downwards while satisfied worms burrow upwards. One possibility is that hungry worms move downwards to feed on abundant but nutrient-poor bacteria growing on plant roots, whereas sated worms can afford to risk moving to the surface in search of more desirable, but less reliable, food sources. Future work could set out to test these explanations.

We show for the first time that the soil nematode *Caenorhabditis elegans* orients to earth-strength magnetic fields. This ability is required for vertical burrowing migrations directionally influenced by their satiation state. The direction and strength of the behavioral response to magnetic fields of wild-type strains isolated around the world correlated with their native magnetic field's inclination, and with the amplitude of the field's vertical (but not its horizontal) component. The AFD sensory neurons respond to earth-strength magnetic fields as observed by calcium imaging, and are necessary for magnetic orientation, and for vertical migrations. Expression of the cyclic nucleotide-gated ion channel, TAX-4, in AFD neurons is necessary for worms to engage in vertical migrations, to orient to artificial magnetic fields, and for the AFD neurons to activate in response to an earth-strength magnetic stimuli.

## Results

### *C. elegans* engages in vertical burrowing migrations

While much is known about *C. elegans* crawling on agar surfaces, in the wild worms likely spend most of their time burrowing through their substrate. After 50 years of *C. elegans* research, studies looking at their burrowing behavior have only recently begun (*Kwon et al., 2013*; *Beron et al., 2015*). Because worms are known to orient to a variety of sensory stimuli that vary with depth in their native soil niches (*Braakhekke et al., 2013*), we hypothesized that burrowing worms engage in vertical migrations like magnetotactic bacteria. To test this, we placed worms in the center of 20-cm long, agar-filled cylinders. Three layers of aluminum foil and a Faraday cage blocked light and electric

fields respectively from penetrating the cylinders. Pipettes were then aligned horizontally in the 'north-south' or the 'east-west' directions, or vertically in the 'up-down' direction in the absence of artificial magnetic fields (*Figure 1A*). Directional preference during burrowing was quantified with a burrowing index computed as the difference between the number of worms reaching either side divided by the total number of worms reaching both sides. We found that when starved, the wild-type lab strain, N2, originally from Bristol, England (*Dougherty and Calhoun, 1948*) preferentially migrated down in vertically oriented cylinders, but did not show a burrowing preference when cylinders were arranged horizontally (*Figure 1B*).

Most animals determine the up and down direction by sensing the gravitational field of the earth (e.g., protozoans: *Roberts, 2010*; crustaceans: *Cohen, 1955*; vertebrates: *Popper and Lu, 2000*). Alternatively, magnetotactic bacteria have been shown to use the earth's magnetic field to migrate up or down within the water column (*Blakemore, 1975*). To help distinguish magnetotactic *vs* gravitactic mechanisms we built a magnetic coil system capable of producing homogeneous magnetic fields of any desired 3D orientation (*Figure 1—figure supplement 1*). Our magnetic coil system is comprised of three independently-powered, orthogonal Merritt coil systems that allow the generation of magnetic fields of up to 3× earth strength (*Figure 1—figure supplement 1*, *Merritt et al., 1983*). Within the 1-m³ coil system, a smaller 20-cm² Faraday cage, made with copper fabric, protects the test volume from the electric field that is concomitantly created alongside the magnetic field. Though often ignored in studies on animal magnetic orientation, this precaution was necessary because *C. elegans* and other animals exhibit strong behavioral responses to electric fields (*Gabel et al., 2007*;

*Manière et al., 2011*). We repeated our vertical burrowing assay with an artificial magnetic field of earth-strength oriented opposite to the local earth's magnetic field (i.e., magnetic north pointing up rather than the natural orientation where magnetic north points down, *Figure 1C*). Under these conditions, we expected that worms responding to gravitational cues would continue to migrate down, while worms responding to magnetic cues would reverse their direction and now migrate up. Consistent with magnetic stimuli dictating vertical migration we found that starved N2 strain worms reversed their burrowing behavior and migrated up (*Figure 1B* red bar).

## *C. elegans* orients to magnetic fields of earth strength in a satiation dependent manner

Our above results indicated that *C. elegans* might be able to detect and orient to magnetic fields of earth strength. To further investigate how worms respond to magnetic fields we placed them at the center of an agar plate and in turn placed this at the center of our 1-m³ Merritt coil system (*Figure 1—figure supplement 1*; *Figure 2A,B*). We placed an anesthetic (NaN₃) around the circumference of the plate, which allowed us to immobilize and tally worms after they arrived at the plate's periphery. To determine if the coil system generated an unwanted temperature gradient, we measured temperatures across the assay plate in response to a magnetic field (*Figure 2—figure supplement 1A,B*). Temperature gradients between the assay's start position (at the center of the plate) and the finish position (at its edge) were negligible

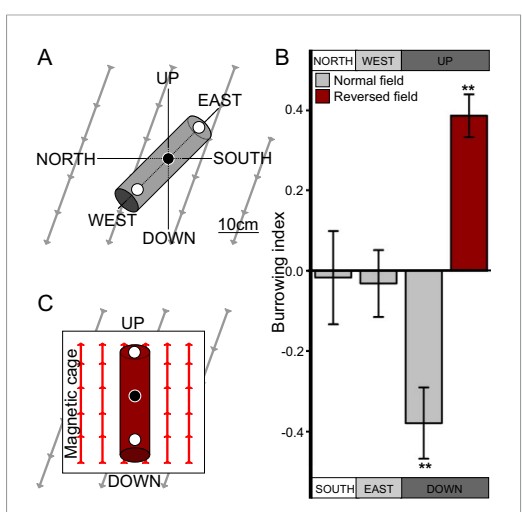

**Figure 1**. *C. elegans* engages in vertical migrations whose direction depends on satiation state. (**A**) To determine if *C. elegans* engaged in burrowing migrations we injected worms into agar-filled pipettes aligned horizontally (east-west and north-south), or vertically (up-down). Alternatively, we disrupted the local magnetic field around vertical pipettes (where magnetic north is down), by reversing the local field polarity (thus making magnetic north up) with a magnetic coil system. (**B**) Only worms in pipettes aligned vertically displayed burrowing bias, preferentially migrating down unless the local magnetic field polarity was reversed (red bar) with the help of a magnetic coil system (**C**).

The following figure supplement is available for figure 1:

**Figure supplement 1**. Merritt coil system for 3D control of magnetic fields.

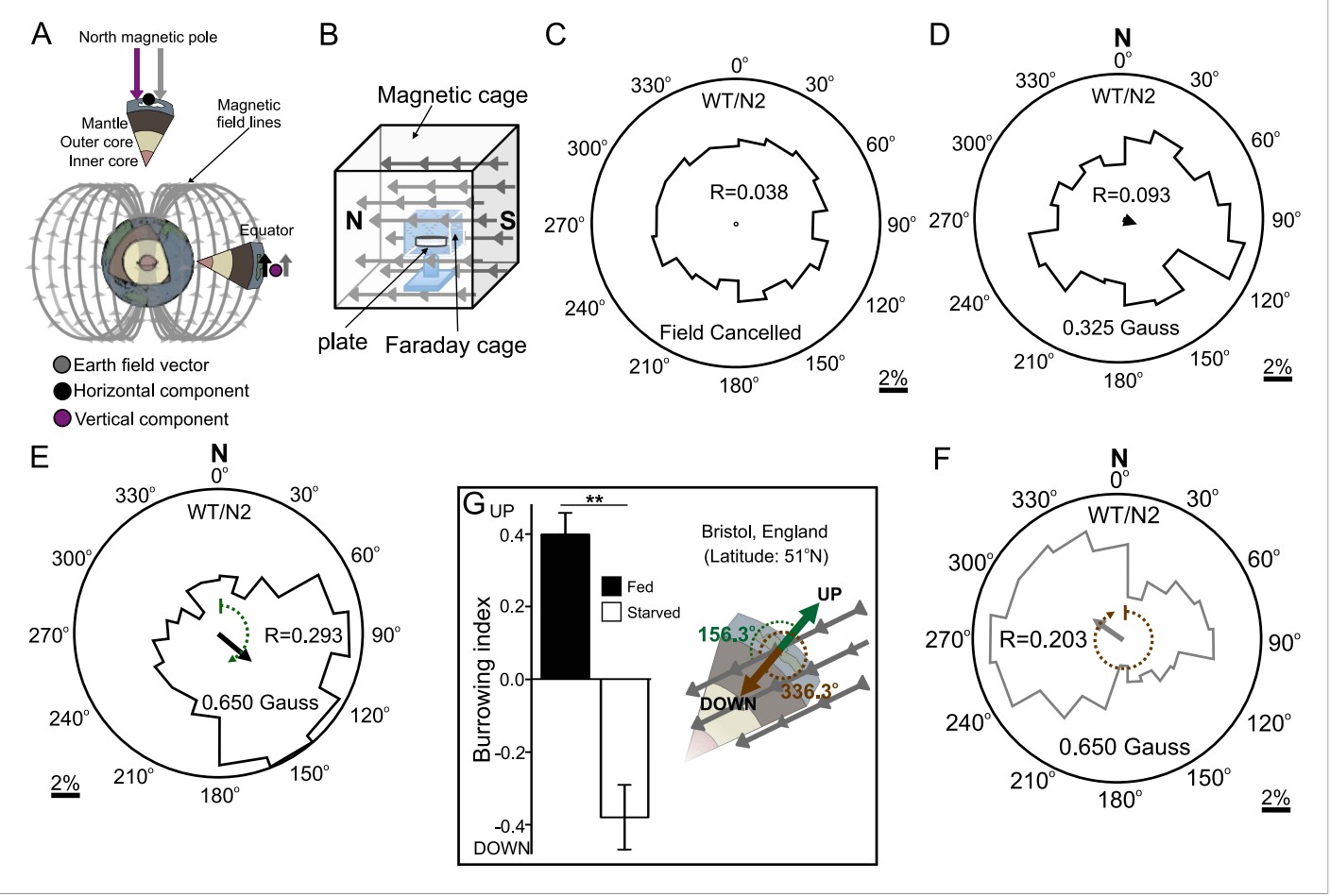

**Figure 2**. Preferred magnetotaxis orientation to a spatially uniform, earth-strength field depends on satiation state, and local field properties. (**A**) The core of the earth generates a magnetic field that bisects the ground at different angles across the planet. The vertical component is strongest at the poles and weakest near the equator. (**B**) We constructed a Faraday cage within three orthogonal magnetic coil systems to test the response of worms under earth-like magnetic conditions. Circular histograms show the average percentage of worms migrating in each of 18 20-degree-wide headings. The radius of each circle represents 10% of the tested population and the mean heading vector arrow would be as long as the radius if every worm converged on one heading, and have zero length if worms distributed randomly around the plate (see *Figure 1—figure supplement 1B* for explanation of circular plots). When the magnetic field around them was cancelled *C. elegans* migrated randomly (**C**). (**D**) Worms migrated at an angle to an imposed field when its amplitude was half maximum earth strength (0.325 Gauss). (**E**) When exposed to a field equaling maximum earth strength (0.625 Gauss), worms showed stronger orientation. (**F**) Starving the worms for 30 min resulted in animals migrating in the opposite direction to their heading while fed. (**G**) Burrowing worms mirrored the magnetic coil results with fed worms preferentially burrowing up while starved worms preferentially migrating down. For the standard lab strain N2 (native to Bristol, England) the virtual up and down direction is represented in the circular plots by a green and brown dashed arch and was not found to vary significantly from the mean heading angle of fed and starved worms respectively. For the magnetic coil assays migration along the imposed field would translate animals towards the 0°/N signs.

The following figure supplement is available for figure 2:

**Figure supplement 1**. Testing the presence of temperature gradients.

across time, and did not vary significantly whether we imposed a magnetic field of one earth strength, or if we cancelled out the earth's field by imposing a field of equal strength but opposite direction (two-way repeated measures ANOVA, N = 5, p = 0.123).

First, as a control, we asked how worms respond when the earth's magnetic field is cancelled. We accomplished this within the test volume of the magnetic coil system using a magnetic field of equal strength and orientation to the field of the earth, but with opposite direction. Worms in this regiment experienced a net magnetic field of 0.000 Gauss in three dimensions. Under this condition

animals migrated randomly, distributing evenly around the circumference of the assay plate (*Figure 2C*). We next produced a homogeneous magnetic field of 0.325 Gauss (corresponding to half of earth's maximum field intensity) directed across the assay plate. Worms assayed this way showed a biased distribution directed ~120° to the imposed magnetic vector (*Figure 2D*). Increasing the field strength to match the earth's maximum field strength (0.650 Gauss) resulted in worms migrating approximately the same direction (132°) to the imposed vector (*Figure 2E*). Surprisingly, if worms were allowed to starve for just 30 min, they reversed their migratory distribution by ~180°, now migrating at 305° relative to the field vector (*Figure 2F*). These results demonstrate that *C. elegans* does not migrate simply toward magnetic north like magnetotactic bacteria (*Frankel et al., 2006*); rather, they display a preference to migrate at particular angles relative to magnetic north that depend on feeding state. Similar plasticity for opposite migration preferences in *C. elegans* has been documented for other sensory modalities (e.g. *Bretscher et al., 2008*; *Russell et al., 2014*).

Do these seemingly arbitrary migratory angles serve a relevant purpose in the worm's soil niche? As mentioned earlier, the standard wild-type *C. elegans* lab strain (N2) was originally isolated in Bristol, England and cryogenically preserved there for distribution and study around the world. We turned to available geomagnetic data from NOAA to determine whether these migratory angles related to the earth's magnetic field in England (*Maus et al., 2009*). In Bristol, the earth's magnetic vector enters the ground (north pointing down) at approximately 66° of inclination (*Figure 2G*). Thus, in Bristol, to optimally orient upward, a worm would need to migrate 156° to the magnetic field penetrating the earth (green arc in *Figure 2E,G*); to optimally orient downward, a worm would need to migrate 336° to the magnetic field (brown arc in *Figure 2F,G*). To determine whether the preferred migratory angles of starved and well-fed worms in our magnetic coil system assay matched these directions we performed a V test (*Batschelet, 1981*). We found that the mean heading of fed worms (132°, N = 1268 animals) did not differ significantly from the upward direction for England (156.3°). Likewise, the mean heading of starved worms (304.6°, N = 1079 animals) did not differ significantly from the downward direction for England (336.3°). Please refer to *Supplementary file 1A* through 1e for descriptive and analytical statistics for all the data presented in this study.

These results predicted that worms use the earth's magnetic field to migrate at angles to the vector that would translate them up if they are fed, or down if they are starved. To test this hypothesis, we placed well-fed or starved worms in vertically arranged agar-filled pipettes away from artificial magnetic and electric fields as before. We found that, consistent with this idea, starved worms preferentially migrated down while well-fed worms migrated up (*Figure 2G*). These results are parsimonious with *C. elegans* directing its vertical burrowing behavior by using the earth's geomagnetic field. Soil nematodes feed on bacteria associated with rotting fruit on the soil surface (*Félix and Braendle, 2010*) and on root rhizobacteria deep in the soil (*Horiuchi et al., 2005*). Vertical migrations could be associated with travel between these distinct food sources.

## Natural variation in magnetic orientation relates to geomagnetic inclination

Like magnetotactic bacteria, *C. elegans* has been isolated across the world. Magnetotactic bacteria from different hemispheres migrate in opposite directions to the field vector. Bacteria that inhabit the northern hemisphere (where magnetic north points down) are termed north-seeking magnetotactic bacteria, while those inhabiting the southern hemisphere (where magnetic south points down) are termed south-seeking magnetotactic bacteria (*Frankel et al., 2006*). The distribution of different wild-type *C. elegans* isolates from around the world with distinct magnetic environments affords us a valuable opportunity to investigate how animals in magnetically distinct environments respond to magnetic fields. We therefore repeated our magnetic coil system and burrowing assays with wild-type *C. elegans* worms isolated from Adelaide (Australia) where the magnetic field of the earth is similar in strength and angle to that in England but differs in the key respect of having the opposite polarity (*Figure 3A*). Unlike British worms, we found that Australian worms placed in a plate within our magnetic coil system migrated to an earth-strength field at 302.5° if fed, and 117.4° if starved. While oriented oppositely in preference from the angles displayed by the British N2 strain, these angles were similar in that they would also result in upward translation in Australia for fed animals and downward translation for starved ones (*Figure 3B,C* respectively). To test if this response to an imposed artificial magnetic field reflected the migratory burrowing preference of worm in a natural magnetic field, we compared the burrowing behavior of Australian worms to that of the British strain.

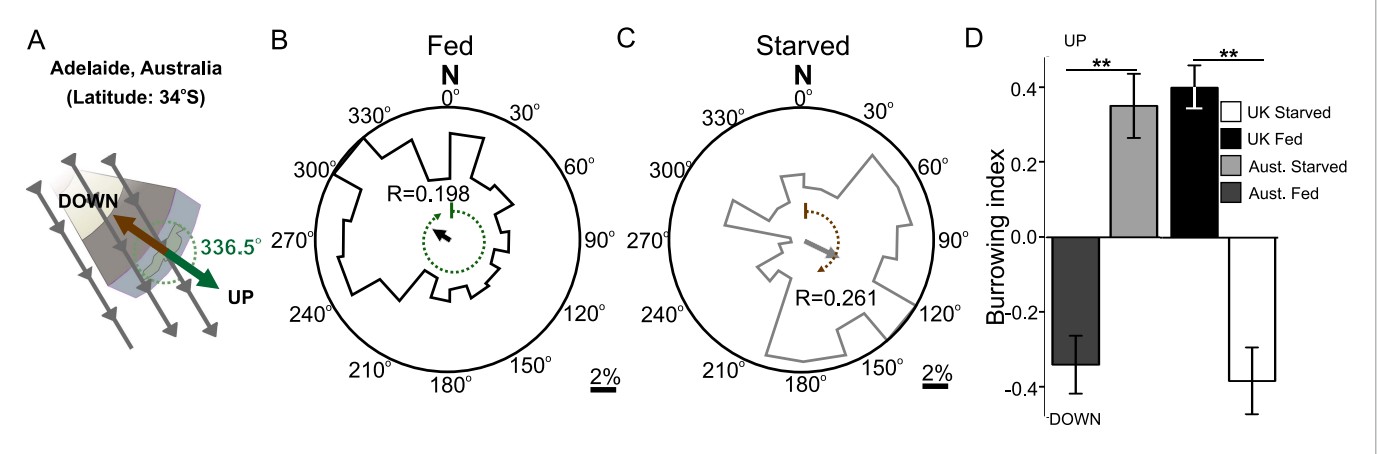

**Figure 3**. Magnetic orientation varies with satiation state and local field properties. To investigate if worms from distinct locations around the world displayed different magnetic orientations we tested *C. elegans* isolated from Adelaide (Australia) where the magnetic field is similar to that of the lab strain (Bristol, England) in strength and inclination but opposite in polarity (**A**). Worms from Australia showed a magnetotactic response reversed from the British strain. Plots for well-fed (**B**) and starved (**C**) worms are shown and the local angle relative to the up and down direction are shown as green and brown dashed arches respectively. For each population, the radius of the circle represents 10% of the animals. The histograms show the percent of the worms that migrated in each of 18 20-degree headings. The mean heading vector shows the average direction of the animals and is equal to zero if all animals migrated randomly, and to the circle radius if all animals migrate on a single heading. (**D**) We compared the burrowing preference of fed and starved British and Australian worms placed in the local (Texas) magnetic field and found that consistent with our magnetic cage experiments both strains migrated in opposite directions.

Paralleling our magnetic coil system results, we found that in our lab (located in Texas, USA) Australian worms migrated down when well-fed, and they migrated up when starved in the burrowing assay (*Figure 3D*). Identical results were found for Hawaiian worms which migrated at angles that would optimally orient them up and down when satiated and starved in Hawaiian (*Supplementary file 1B*). Overall, these results suggest that unlike magnetotactic bacteria, which follow magnetic field lines during their migrations, worms migrate at angles to the imposed field that would result in optimal vertical translation in their native locations.

## Magnetotactic ability correlates with global field properties

The results of our magnetic coil and burrowing experiments suggested that worms use the local magnetic field to guide vertical migrations. Unfortunately these experiments are limited to a few assays at the time, preventing their use in larger-scale behavioral screens. To mitigate this shortcoming, we developed a new assay using strong rare-earth magnets to quickly assess the ability of different strains to respond to imposed magnetic fields (*Figure 4A*). This assay allowed us to run many assays at the same time. Briefly, worms were placed at the center of an assay plate and allowed to migrate freely (*Figure 4A*, and 'Materials and methods'). A magnet was then placed above one of two equidistant 'goal' areas. Magnetotactic performance was quantified with a magnetotaxis index computed as the difference between the number of worms reaching either goal divided by the number of worms reaching both goals. We found that when no magnet was present, worms distributed evenly between these two goals. However, if the magnet was present, worms preferentially migrated toward it (*Figure 4—figure supplement 1*, *Supplementary file 1A*). To ensure the presence of the magnet did not introduce an unwanted thermal gradient, we recorded the temperature difference between goals in the presence and absence of a magnet and found that the two treatments did not significantly differ from each other (*Figure 2—figure supplement 1C,D*). We used this assay to compare the ability of different strains to detect and migrate in a biased way in the presence of strong magnetic field. We first turned our attention to many wild *C. elegans* strains isolated from different locations across the world.

The magnetic field of the earth varies greatly around the world (*Maus et al., 2009*). If *C. elegans* uses the magnetic field for vertical migrations, what happens near the equator where the vertical component of the earth's field is weakest? The global heterogeneity in field characteristics made us

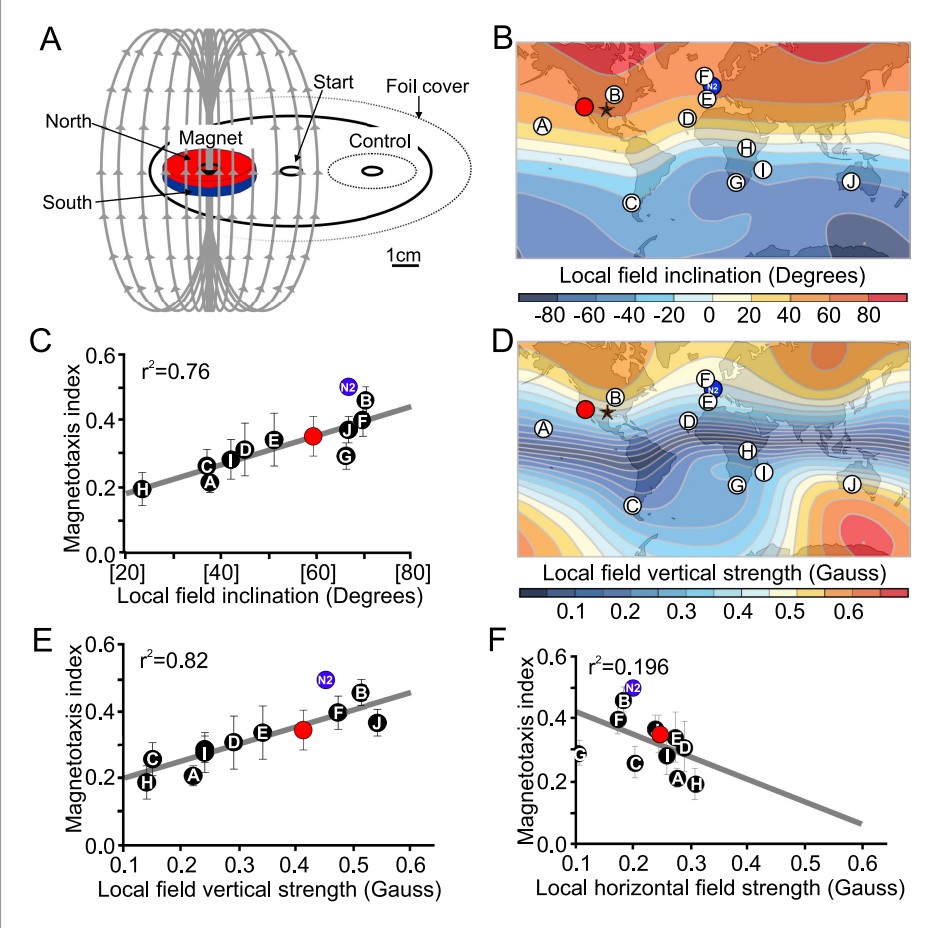

**Figure 4**. Magnetotactic variability between wild *C. elegans* isolates result from differences in local magnetic field properties. (**A**) We developed a novel assay to rapidly assess the ability of worms to detect and orient to magnetic fields. Worms placed at the center of a test plate were allowed to migrate freely toward or away from a magnet. The number of animals by the magnet *M,* or by a control area *C* were compared and used to calculate a magnetotaxis index: MI = (M − C)/(M + C). Wild-type *C. elegans* have been isolated across the planet at locations with diverse local magnetic fields. (**B**) Earth's magnetic field inclination map plotted from data obtained from NOAA (***Maus et al., 2009***) showing the isolation location for twelve wild-type strains of *C. elegans* used in this study (circles). (**C**) The ability of these wild isolates to magnetotax in our magnet assay strongly correlated with the inclination of the magnetic field at their origin. We used the white ten isolates to compute the correlation between these variables. This correlation was able to predict the magnetotaxis index of an additional strain obtained from California (red circle). (**D**) Map of the vertical component of the earth's magnetic field (***Maus et al., 2009***). (**E**) Performance in the magnet assay was even more correlated with the vertical component of the earth's magnetic field. However, the horizontal component of the magnetic field (**F**) showed no correlation with the magnetotaxis index of the wild isolates. The blue circle represents the lab strain (N2) from England. All assays conducted at location indicated by the lone star. All values reported are means. Error bars represent S.E.M.

The following figure supplement is available for figure 4:

**Figure supplement 1**. A new assay for testing magnetotactic ability.

wonder if selection pressure for magnetosensation ability may drop off nearest the equator where the vertical component of the magnetic field is at its weakest. To test this, we used our magnet assay (***Figure 4A***) on wild-type populations isolated from ten locations across the planet where the local magnetic field varies in inclination and vertical strength (***Figure 4B,D***, ***Maus et al., 2009***). We found that the ability of different worm populations to orient to an artificial magnetic field was strongly correlated with the inclination (***Figure 4B,C***) and vertical strength (***Figure 4D,E***) of the magnetic field

at their native sites. The horizontal component of the field, however, was a poor predictor of this magnetotactic ability (*Figure 4F*). The strong correlation between magnetotaxis performance, local field inclination, and vertical strength allowed us to successfully predict the magnetotaxis index of an additional wild-type isolate from California, USA (*Figure 4* red circle in panels B-E, *Supplementary file 1C*). Wild-type isolates from equatorial locations where the magnitude of the vertical component was close (or below) 0.2 Gauss, were either unable or barely able to magnetotax (*Supplementary file 1A*). These results are consistent with local adaptations to global magnetic field variations, and could perhaps be used to model how other species may respond to temporal field variations (such as magnetic polar drift or field reversals, *Cox et al., 1964*). We conclude from these results that like many animals (*Johnsen and Lohmann, 2005*) *C. elegans* can use the magnetic field's polarity and inclination to guide its migrations. Having determined that *C. elegans* orients to magnetic fields, we turned our magnet assay to next investigate the cellular and molecular underpinnings of this fascinating behavior.

## AFD sensory neurons are necessary for magnetic orientation

To investigate the neuromolecular substrates for magnetosensation, we tested mutants with deficiencies in a variety of previously characterized sensory pathways. Mutants with severe defects in some sensory modalities displayed normal or nearly normal magnetic orientation (*Figure 5*). These included worms deficient in the touch-form of mechanosensation (*mec-10*, *Arnadottir et al., 2011*), light detection (*lite-1*, *Edwards et al., 2008*), taste (*che-1*, *Uchida et al., 2003*), and oxygen sensation (*gcy-33*, *Zimmer et al., 2009*). However, we also found mutants that were significantly impaired in magnetotaxis. This group comprised worms with mutations in genes co-expressed in a single sensory neuron pair called AFD, first implicated in thermosensation (*Mori, 1999*). These included two independent mutant alleles of *ttx-1*, important for AFD differentiation, and the triple mutant lacking guanylyl cyclases, *gcy-23, gcy-8,* and *gcy-18*, which together are critical for AFD function. Furthermore, we identified a set of transduction mutants that failed to perform magnetic orientation. These included two independent mutant alleles of each *tax-4* and *tax-2* genes. These encode subunits of a cGMP-gated ion channel already implicated in sensory transduction in many sensory neurons, including AFD (*Komatsu et al., 1996*).

To test the requirement of the AFD neuron pair in magnetosensation, we genetically ablated them via cell-specific expression of a transgene for a human cell-death caspase. One advantage of this technique is that only a fraction of individual worms will inherit the artificial chromosome carrying the transgene. This allowed us to compare the performance of sister worms grown and tested together under identical conditions and only differing in having or not said transgene. After each assay, individual worms with genetically ablated neurons were distinguished from their unaffected sisters by the co-expression of a fluorescent transgene reporter. We found that worms lacking the AFD sensory neurons failed to orient to an artificial magnetic field, while their unaffected sisters oriented normally (*Figure 6A*, *Supplementary file 1A*). This could not be explained by non-specific defects, because these worms could move and orient normally to olfactory stimuli (*Figure 6—figure supplement 1*). Similarly ablating nearby sensory neuron pairs ASE and AWC had no effect on magnetotaxis. The sensory ending of the AFD neurons consists of dozens of villi arranged anterior-to-posterior (in an antenna-like formation) imbedded inside glial cells (*Perkins et al., 1986*; *Doroquez et al., 2014*). Genetic ablation of the glia surrounding these structures, results in worms with viable AFD

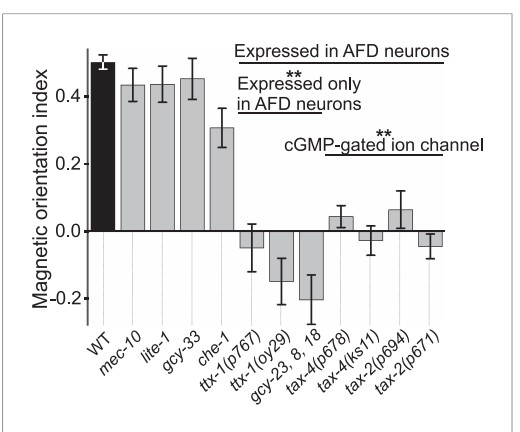

**Figure 5**. Magnetotaxis requires intact AFD sensory neurons. We used our magnet assay to test a large number of sensory mutants. Mutations that impair the mechano- (*mec-10*), light- (*lite-1*), oxygen- (*gcy-33*), and taste- (*che-1*) sensory pathways spared magnetotaxis, while mutations in genes specifically required for AFD sensory neurons (*ttx-1* and *gcy-23,-8,-18*) abolished magnetotaxis. Mutations that impair the cGMP-gated ion channel TAX-4/TAX-2 that are expressed in the AFD sensory neurons (and other cells) similarly prevented magnetotaxis.

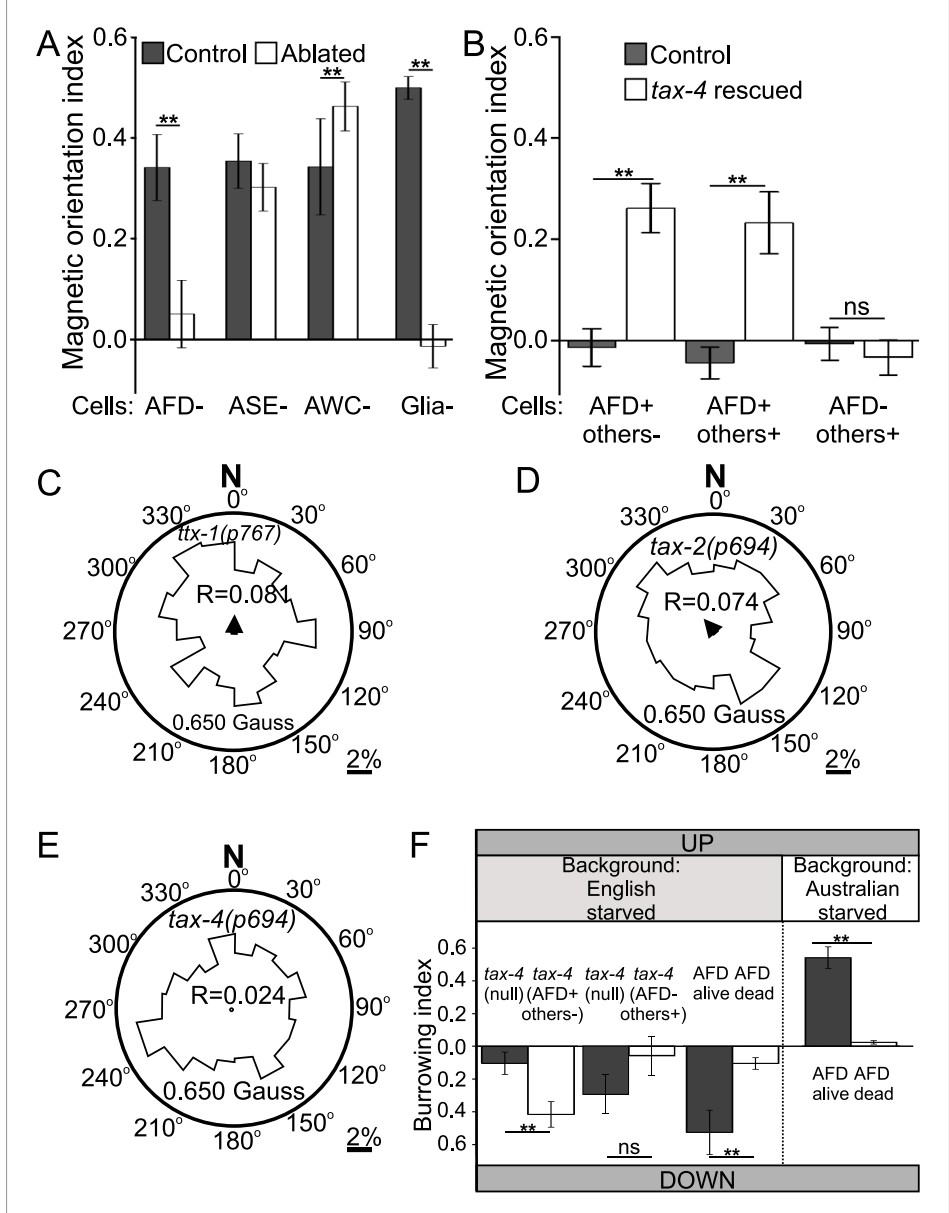

**Figure 6**. Geomagnetotaxis requires the TAX-4/TAX-2 cGMP-gated ion channel in the AFD sensory neurons. (**A**) Genetic ablation of the AFD neurons (or their sensory villi via ablation of amphid glial cells) prevented magnetotaxis. However, ablation of adjacent sensory neurons (ASE and AWC) did not impair this behavior. (**B**) Genetic rescue of the cGMP-gated ion channel TAX-4 via cDNA specifically in the AFD neurons, or via genomic DNA in additional *tax-4*-expressing neurons was sufficient to restore magnetotactic ability (white bars) compared to their *tax-4*-mutant background controls (grey bars). However, rescue of tax-4 expressing neurons that excluded the AFD neurons failed to restore magnetotactic behavior. We retested some of the mutants impaired in the magnet assay in the magnetic coil system under earth-like fields. Impairment of the AFD neurons by mutations in the *ttx-1* (**C**), *tax-2* (**D**), or *tax-4* (**E**) genes resulted in worms that failed to orient to magnetic fields of earth strength (0.625 Gauss). Migration along the imposed field would translate animals towards the 0°/N mark. (**F**) Genetic manipulations that impaired (or rescued) magnetotaxis had a similar effect on geomagnetotaxis of vertically burrowing worms. Starved British worms lacking the *tax-4* gene failed to burrow down. However, control sister worms with the *tax-4* gene rescued specifically in the AFD neurons (AFD+ others-) were able to burrow down. Conversely, starved British worms lacking the AFD neurons (AFD dead) failed to migrate down, while control sister

*Figure 6. continued on next page*

*Figure 6. Continued*

worms (AFD alive) migrated down. Ablation of AFD in Australian worms similarly abolished geomagnetotaxis.
* p < 0.05, ** p < 0.001. All values reported are means, and error bars represent S.E.M.

The following figure supplement is available for figure 6:

**Figure supplement 1**. Genetic ablation of AFD does not impair chemotaxis.

---

neurons but lacking villi (*Bacaj et al., 2008*). These worms were unable to orient to artificial magnetic fields (*Figure 6A*). This supports the idea that the villi may be the site of magneto-transduction (and/or that the glia themselves contribute to this sense). Taken together, our results demonstrate that the AFD sensory neurons are required for magnetotaxis.

## TAX-4 cGMP-gated channel mediates magnetic orientation in the AFD neurons

Many sensory neurons in *C. elegans* require the TAX-4 cGMP-channel for sensory transduction (*Komatsu et al., 1996*). To determine if TAX-4 function in the AFD neurons was sufficient for magnetic orientation, we selectively rescued expression of *tax-4* in the AFDs neurons in a *tax-4* mutant background. Specific rescue of TAX-4 in AFD neurons was sufficient to partially restore the ability of *tax-4(null)* mutant worms to orient to an artificial magnetic field (*Figure 6B*, AFD+ others-). To investigate the possibility that TAX-4 may also mediate magnetic orientation through additional neurons we further rescued TAX-4 in all *tax-4*-expressing neurons by using its endogenous promoter and regulatory elements. However, this did not result in an increased rescue (*Figure 6B*, AFD+ others+). To test if *tax-4* contributed to magnetotaxis through any other neuron asides from AFD, we tested *tax-4* mutants where this gene was rescued in all *tax-4*-expressing neurons except for the AFD neurons (gift from Dr R Baumeister). While rescuing *tax-4* in all but AFD neurons resulted in a rescue of the ability of these animals to orient to chemical stimuli (*Supplementary file 1A*), these animals remained unable to orient to magnetic fields (*Figure 6B*, AFD- others+).

These results support the hypothesis that the cGMP-gated ion channel TAX-4 plays an important role in the AFD sensory neurons for orientation to magnetic fields. To confirm the relevance of these findings in a more natural magnetic assay, we retested selected mutants under earth-like fields (in our coil system) and found similar results (*Figure 6C–E*). The results above were obtained for worms orienting to artificial magnetic fields. To determine if these results generalized to the ability of worms to engage in vertical migrations we tested selected strains in our vertical burrowing assay (without artificial magnetic field). Consistent with our observations in the magnet and in the magnetic coil assays, *tax-4* mutant worms did not show preferential vertical migration unless the gene was selectively rescued in the AFD neurons (*Figure 6F*). Starved British and Australian wild-type isolates lacking the AFD neurons similarly failed to engage in biased vertical migrations, although their sisters not carrying the transgene (used to kill AFD) remained able to migrate down or up respectively (*Figure 6F*).

## The AFD neurons respond to earth-strength magnetic fields

To determine whether the AFD neurons are directly responsive to magnetic fields, we measured the fluorescence of a genetically encoded calcium indicator, GCaMP3, in fully immobilized worms (*Figure 7A*, and *Figure 7—figure supplement 1A*). After recording baseline activity (*Figure 7B*), we exposed mechanically immobilized worms to an 8-s, 65-Gauss (100× earth) rotating (2 Hz) magnetic stimulus (see 'Materials and methods' for details). We observed a transient increase in the average brightness of the AFD neurons (*Figure 7C*). Successive stimuli consistently produced a reduced response (*Figure 7D*). We observed a similar response when the magnetic stimulus was decreased to 10× and 1× earth stimuli (6.5 and 0.65 Gauss respectively, *Figure 7E,F*). To help determine if the AFD neurons themselves are magnetosensitive, and not just synaptically downstream from 'real' magnetoreceptive neuron(s), we measured AFD calcium responses in worms impaired in rapid and dense-core synaptic transmission (*unc-13* and *unc-31* mutant strains, *Ahmed et al., 1992*; *Ann et al., 1997*). In the absence of chemical synaptic or neuromodulatory inputs, the AFD neurons continued to respond to magnetic fields (*Figure 7G,H*). Qualitatively similar results (but higher in amplitude) were

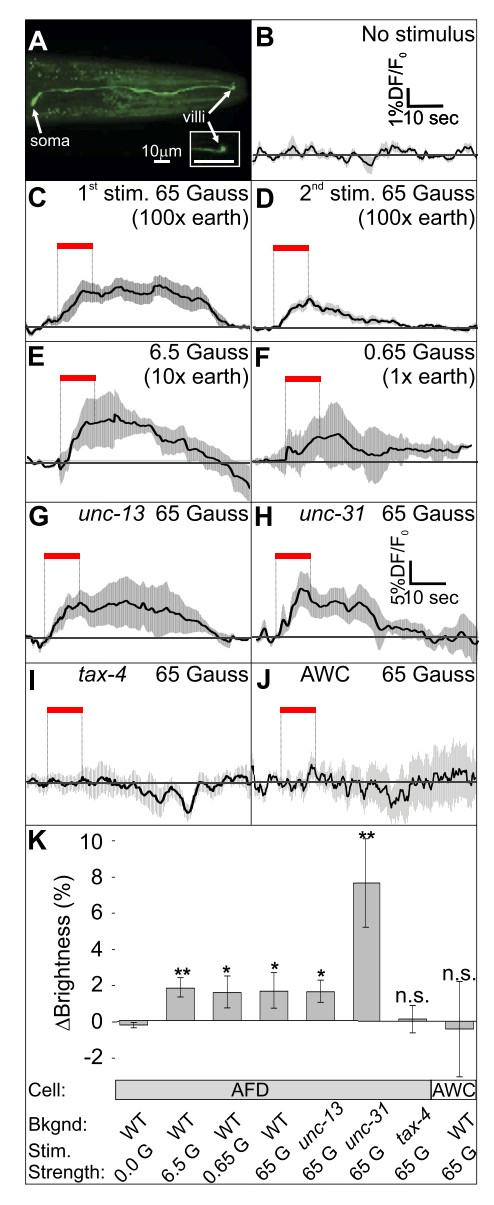

**Figure 7**. The AFD sensory neurons respond to magnetic stimuli. (**A**) Calcium activity indicator GCaMP3 in the AFD neurons. (**B**) In the absence of a magnetic stimulus the soma of AFD neurons rests at baseline. Exposing restrained worms to a sinusoidal 65 Gauss (100× earth strength) magnetic stimulus caused the soma of the AFD neurons to transiently increase brightness by 2% above baseline in response to the first stimulus (**C**), and ~1% in response to subsequent stimuli (**D**). The AFD neurons responded when the magnetic stimuli was reduced to 6.5 (**E**) and 0.65 Gauss (**F**, earth strength). The AFD magnetic response remained even in synaptic mutants (**G**: *unc-13* and **H**: *unc-31*) that render these cells synaptically isolated from other neurons. (**I**) Animals lacking a functional copy of the *tax-4* gene did not show an increase in brightness in response to a magnetic stimulus. (**J**) A 65 Gauss stimulus failed to elicit a response in neighboring sensory

observed for worms that were partially restrained (*Figure 7—figure supplement 1B*). Magnetic-induced calcium responses in AFD were not observed in a *tax-4* mutant background, suggesting that this requires $Ca^{2+}$ entering the TAX-4 cGMP-gated ion channel (*Figure 7I*, *Figure 7—figure supplement 1B*). Responses were also not observed in an adjacent sensory neuron pair AWC (*Figure 7J* and *Figure 7—figure supplement 1B*). To quantitatively compare the magnetosensory response of AFD for different conditions and mutant backgrounds, we plotted the average GCamp3.0 intensity during the final 4 s of the magnetic stimulus relative to a 4-s baseline before presentation of the stimulus (for the no-stimulus control we used the same time window as for the other recordings). We found that the change in brightness was significantly greater than control for all test conditions except in the case of *tax-4* mutant background (*Figure 7K*). Our imaging results provide physiological evidence that the AFD sensory neurons respond to magnetosensory stimuli relevant to geomagnetic orientation.

## Discussion

Here we provide the first behavioral and physiological evidence for magnetosensory neurons. Nematodes unexpectedly appear to use the AFD neurons to orient to earth-strength magnetic fields. *C. elegans* guides its vertical migrations using the geomagnetic field and adjusts preference for up or down depending on their satiation state. Population variability in magnetotactic ability correlates with global field properties: with worms from locations where the field is strong and vertical outperforming those where the field is weak and more horizontal. The AFD sensory neuron pair is necessary for magnetic orientation and for vertical migrations. Similarly, a cGMP-gated ion channel in the AFD neurons, TAX-4, is also necessary and sufficient for these behaviors. The role of the AFD sensory neurons in magneto-transduction is supported by their ability to respond to magnetic fields even in the absence of synaptic inputs.

### Cellular and molecular substrates of magnetotransduction

There are many possible ways in which the AFD neurons may play a role in magnetic orientation. The magnetosensory AFD neurons also respond to temperature (*Mori, 1999*), $CO_2$ (*Bretscher et al., 2008*; *2011*), and moisture (*Russell et al., 2014*) gradients in a satiation-dependent

*Figure 7. Continued*

neuron AWC. (**K**) The average soma brightness for the final 4 s prior to stimulus, and the final 4 s of the stimulus were compared. While the 'no-stimulus', the '*tax-4*', and the 'AWC' conditions resulted in no significant brightness change, all other test conditions produced a significant increase in AFD brightness above baseline. Change in relative fluorescence key for panels **B**–**J** depicted in **B** with the exception of panel **H** which has its own key. N = 11 for **B**–**D**; 6 for **E**–**H**; 14 for **I**; and 7 for **K**.

The following figure supplement is available for figure 7:

**Figure supplement 1**. Measuring AFD calcium activity in partially and fully restrained worms.

manner. All of these parameters vary with depth in the soil (*Jassal et al., 2005*) supporting the role of the AFD neurons in vertical burrowing migrations. Because *C. elegans* performs magnetotaxis in darkness (assays were wrapped in 3-layers of foil), it is possible that it detects fields with nano-scale compasses made of biological magnetic material previously described in *C. elegans* (*Cranfield et al., 2004*), rather than by a light-dependent mechanism, although this study did not investigate this possibility. However, based on our behavioral, mutant, transgenic, and physiological analyses we hypothesize magnetic particles, perhaps such as those found in magnetotactic bacteria (*Frankel et al., 2006*) and previously reported in *C. elegans*, may (either directly or indirectly) be associated with the anterior and posterior-directed microvilli of the AFD neurons. Magnetic stimulation of these structures could lead to activation of unspecified guanylyl cyclases (such as GCY-8, GCY-18, or GCY-23), in turn activating the TAX-4 channel and resulting in $Na^+$ and $Ca^{2+}$ influx and cell depolarization. These findings could represent an intriguing lead into the putative magnetotransduction mechanism of the AFD neurons. It will be intriguing to investigate whether the diverse range of other magnetotactic animals employ magnetosensitive neurons with analogous morphology and transduction mechanisms as the AFD neurons.

## Use of the earth's magnetic field in vertical migrations

The magnetic field of the earth provides reliable directional and positional information to organisms capable of its detection. Aside from magnetotactic bacteria, magnetic orientation has been largely observed in animals that migrate horizontally (*Johnsen and Lohmann, 2005*). Our finding that vertical migrations by an animal may also be guided by this sensory modality opens a new niche for the study of magnetic navigation. Magnetotactic bacteria passively migrate along field lines: with south-seeking bacteria swimming down, and north-seeking bacteria swimming up in the southern and northern hemispheres respectively (*Frankel et al., 2006*). Unlike bacteria, *C. elegans* does not follow the field vector but rather migrates at an angle that appears to maximize its vertical translation. The difference in migration angles and in response amplitude between wild-type isolates from around the world suggests that this sensory modality is under considerable selective pressure and will be the subject of future studies.

Our findings that the direction of vertical migrations could be reversed by an imposed magnetic field and that wild-type populations of worms from opposite hemispheres displayed opposite vertical migration preference strongly suggests that *C. elegans* relies on the geomagnetic field rather than gravity. Many organisms deduce their vertical orientation by using the earth's gravitational field (gravitaxis). Studies on paramecia suggest that the relative density of the organism against its media is instrumental in gravitaxis (*Kuroda and Kamiya, 1989*). For terrestrial and marine animals, the relative density of their media (air and water respectively) is largely constant. However, for nearly buoyant worms imbedded in a soil matrix, the relative density of their surrounding media is highly variable and may preclude reliance on gravitaxis.

Many magnetotactic bacteria use a mechanism known as 'polar magneto-aerotaxis' where these cells preferentially migrate up or down in chemically stratified water or sediment columns using an single sensory pathway to integrate magnetotaxis and aerotaxis (*Blakemore et al., 1980*; *Frankel et al., 1997*; *Popp et al., 2014*). Like *C. elegans*, polar magneto-aerotactic bacteria from different hemispheres have adapted their polarity preference to match their native environment (*Blakemore, 1975*). It appears that the role of magnetotaxis in *C. elegans*, as in bacteria, may be to increase the efficiency of taxis to other sensory cues by reducing a search problem from three dimensions to one. Unlike bacteria which migrate along the dip line at an angle relative to the vertical direction, however, *C. elegans* appears to align motion at an angle to the magnetic field that would enable a more vertical trajectory in its native environment. Our finding that the strains from England, Australia and Hawaii

each displayed a preferred magnetic orientation preference that matched the geomagnetic field orientation at their source of isolation rather than the one at the experimental site (Texas, USA) suggests a genetic encoding of magnetic orientation preference.

## Magnetic orientation preference correlates with the physiological state attained by the animals

From previous work (*Bretscher et al., 2008*; *Russell et al., 2014*), it is clear that satiation affects the sign of the response to sensory stimuli that the AFD neurons respond to. However, our experiments did not provide evidence to answer why fed worms migrate up and starved worms migrate down. One possibility concerns the vertical stratification of food sources. *C. elegans* eats bacteria growing on rotting fruit on the soil's surface (*Félix and Braendle, 2010*), and also root rhizobacteria deep in the soil (*Horiuchi et al., 2005*). Vertical migrations may direct travel between these segregated food sources following the marginal value theorem (*Carnov, 1976*; *Milward et al., 2011*). Rotting fruit on the surface represents an extremely rich, but transient, food supply. By contrast, rhizobacteria represent a low-quality but stable source of food. Surface populations likely grow exponentially until they exhaust their resources. For a starved worm on the surface, burrowing down would be adaptive because it leads to rhizobacteria in the plant roots. Rhizobacteria, however, represent a lower quality food source. From here, fed worms may venture to emerge in search of better and more plentiful food. Worms on poor diets have been shown to be more likely to abandon the relative safety of their food patch in an attempt to find a higher quality source (*Shtonda and Avery, 2006*). For these worms, an adaptive locomotor strategy would be to burrow up in search of higher quality food sources, as burrowing down would not be likely to result in finding a higher quality food patch. Future experimental studies will distinguish between these and other possibilities by mimicking specific soil conditions.

Many animal species (including other nematodes, *Prot, 1980*) engage in vertical soil migrations (*Price and Benham, 1977*). Therefore magnetic orientation may be more widespread than previously believed. While the scale and nature of magnetosensation make it challenging to study in large animals with extensive ranges, the small size, genetic tractability, and research amenability of *C. elegans* make it an optimal model to begin to unlock potentially conserved cellular-molecular mechanisms by which animals detect and orient to the magnetic field of the earth.

## Materials and methods

### Test location

All worms were raised and tested at The University of Texas at Austin, Texas, USA (30° 20′ N 97° 45′ W) between 2011 and 2014. The local characteristics of the magnetic field of the earth during the duration of the experiments were as follows: Declination 4°35′ to 4°13′(East); Inclination 59°19′ to 59°12′ (Down); Horizontal Intensity 0.245 to 0.244 Gauss; Vertical Intensity 0.413 to 0.410 Gauss (Down); Total Intensity 0.480 to 0.477 Gauss (*Maus et al., 2009*).

### Animals

We conducted over 1200 assays (>61,000 worms), averaging ~48 worms per assay. All behavioral assays were conducted with experimenter blind to genotype of the worms assayed. Because of the multimodal properties of the AFD neurons, assays controlled for many physiological and environmental aspects prior to testing. To ensure worms were in comparable physiological states, all assays were performed on (never starved) day-1 adult hermaphrodite *C. elegans*. Worms were never allowed to overpopulate their plates. To minimize physiological changes due to unsealing of test plates (e.g., altering the $O_2/CO_2$ ratio), worms were tested within 20 min of unsealing their incubation plates. To test worms in comparable satiation states, worms tested under the 'fed' status were assayed within 10 min of being extracted from their bacterial lawn. To test worms in the 'starved' state, we allowed worms to remain suspended in liquid Nematode Growth Media (NGM) for 30 min prior to beginning their run. Incubation temperature was between 19–21°C in standard NGM agar plates seeded with *Escherichia coli* (OP50) lawns (*Brenner, 1974*). Artificial magnetic fields were removed from the vicinity of the worms, and the local field surrounding the worms was determined to be of earth strength and direction with a DC Milligauss Meter Model MGM magnetometer (AlphaLab, Utah, USA). To minimize

novel background mutations, all strains were tested within 3 months of thawing from cryopreserved stocks, with additional re-thaws of fresh samples at 3-month intervals.

## Genetic manipulations

We used the GATEWAY system to generate transgenes for transgenic strains (*Hartley et al., 2000*). Fluorescent reporters were used to identify transgenic individuals (*Pmyo-2::mCherry*, *Pmyo-3::mCherry*, or *Punc-122::GFP*; see *Supplementary file 1E* for specifics). We used the AFD-specific promoter (*Pgcy-8*) to target the AFD neurons (*Inada et al., 2006*). To genetically ablate these neurons, we constructed plasmids containing the human caspase gene *ICE* (gift from V Maricq) and transformed N2 wild-type worms to generate strains JPS264 *vxEx264[Pgcy-8::ICE]*. Identical results were found for two independently derived strains, JPS265 and JPS271, in a N2 background. AFD neurons were also killed in the Australian wild-type isolate AB1 to generate strain JPS545 and JPS546. The ability of this transgene to kill the AFD neurons was assessed by comparing GFP expression in the AFD neurons of worms carrying the ICE construct, with that of worms not carrying it (*Figure 6—figure supplement 1*). To measure intracellular calcium levels in AFD neurons in vivo we expressed GCaMP3 (*Tian et al., 2009*; gift from L Looger) *vxEx316[Pgcy-8::GCaMP3]* in *lite-1(ce314)* worms to generate the strain JPS316. Identical results were found for three independently derived strains, JPS275, JPS294, and JPS315. All of these strains were capable of magnetosensation. We rescued *tax-4* specifically in AFD by constructing plasmid with a wild-type copy of the *tax-4* cDNA (gift from Dr Ikue Mori) expressed in AFDs as described above to generate the strain JPS458 *tax-4(ks11) vxEx458[Pgcy-8::tax-4(+)]*. Identical results were found for the independently derived strain, JPS459. We also rescued with fosmid VRM069cE04 containing genomic region of *tax-4* with its promoter, UTR and endogenous regulatory elements *vxEx458[VRM069cE04]* with strain JPS458. *unc-13* or *unc-31* mutants were crossed with JPS316 males and F$_2$ worms were selected that exhibited GCaMP3 fluorescence and an uncoordinated phenotype to generate strains JPS496 and JPS495 respectively. For details on the construction of additional neuronal ablation strains please refer to the Extended Data section in *Russell et al. (2014)*.

## Magnetic response assays

To determine if worms could sense and respond to magnetic fields, we picked 50 never-starved (day-1) adults from an OP50 bacterial lawn and into a 1-μl drop of liquid NGM. We used the latter to clean the worms off bacteria, and to transfer worms to the center of a 1-day old, 10-cm diameter, chemotaxis-agar assay plate. Equidistant from the worms, we drew 3.5-cm circles on either side and placed 1-μl drops of 1-M NaN$_3$ at the center of these circles to immobilize and count any worm that reached the area (*Figure 4A*). A N42 Neodymium 3.5-cm diameter magnet (K&J Magnetics Inc., Pennsylvania, USA) was placed above one of the circles so that the assay plate was now traversed by a vertical magnetic field gradient that became stronger toward the magnet and weaker away from it. (Note that more commonly found weaker strength magnets produced qualitatively the same behavioral results.) Worms were released from the liquid NGM droplet by wicking excess liquid with filter paper. The total manipulation time (from bacterial lawn to the beginning of the assay) was kept under 10 min to avoid inadvertently starving the worms. Worms released from the liquid NGM became able to freely migrate around the plate. After 30 min we counted the number of worms NaN$_3$-paralyzed in each circle and calculated the magnetic orientation index (MI) as: MI = (M − C)/(M + C). Where M is the number of worms paralyzed within the magnet's circle, C is the number of worms paralyzed within the control circle. We repeated the test a minimum of 10 times for each population (please refer to *Supplementary file 1A,C–E* for the number of assays and worms used in each experiment). The absence of artificial magnetic fields and temperature gradients were empirically determined before each assay with DC Milligauss Meter Model MGM magnetometer (AlphaLab, Utah, USA) and two high accuracy Fisher thermometers accurate to 1/100 of a degree (*Figure 2—figure supplement 1E,F*). Assays were run over multiple days and across a range of temperatures (19–21° Celsius). To ensure that unaccounted gradients in the room did not affect the assays, we ran multiple assays in parallel. We arranged assay plates so that their magnetic gradients were not aligned with one another or with the magnetic field of the earth. In this configuration, the magnetic field on the plate surface ranged between 40 and 2900 Gauss (*Figure 4A*).

To test if worms could perform magnetotaxis in the dark, we wrapped assay plates in heavy-duty aluminum foil at least three layers thick. All burrowing experiments were conducted similarly with

the pipettes wrapped in multiple layers of aluminum foil. All magnetic coil system experiments were conducted in the dark. Additionally, burrowing and magnetic coil system experiments were conducted within an opaque Faraday cage consisting of copper mesh.

## Magnetic coil system assays

In order to test worms under earth-like homogeneous magnetic fields we constructed a triple Merritt coil system (*Merritt et al., 1983*) of 1 m$^3$ in volume capable of generating a homogeneous magnetic field in the central 20 cm$^3$ of the space. A 22 cm$^3$, copper fabric, Faraday cage around the test volume prevented electric fields from interfering with our assays. Each of the three coil systems was orthogonal to the other two (*Figure 1—figure supplement 1*) and was independently powered by Maxtra Adjustable 30V 5A DC power supplies. We used a DC Milligauss Meter Model MGM from Alphalab Inc. (Utah, USA) to measure the magnetic field inside the coil system before and after each experiment. Before the start of each experiment, the system was used to neutralize the magnetic field of the earth within the coil system by creating a field of equal strength and opposite orientation. A single 10-cm diameter, agar-filled plate (*Ward, 1973*), with ~50 worms in its center, was placed in the center of the coil system (*Figure 1—figure supplement 1*). To immobilize and count the worms that reached the plate's edge, we placed a 10-μl ring of 0.1-M NaN$_3$ anesthetic in the agar around circumference of the plate. We next closed the Faraday cage and allowed the worms to migrate freely within the plate with a homogeneous earth-strength (0.325 and 0.650 Gauss) magnetic field aligned with the plane of the assay plate. Alternatively, we allowed worms to migrate in plates when the effective magnetic field inside the coil system was 0.000 Gauss (earth-neutralized). After an hour, the angle at which each worm had migrated with respect to the imposed field vector was recorded. To ensure that the worms were responding to the generated magnetic field, and not to some unknown gradient in the room, all assays were run in darkness with the direction of the imposed magnetic field, and also the orientation of the magnetic coil system was varied between trials with respect to the room and the earth's magnetic north. All experiments were conducted blind with experimenter unaware of strain genotype.

In addition to magnetic and electric fields, the wires of coil system also produce a small degree of heat. To test for the presence of unwanted temperature gradients we used two high accuracy (0.01°C) thermometers (Fisher Scientific, New Hampshire) to record the temperature at the center of the plates (where the worms begin the assay) and at the edge of the plates (where they complete the assay). Please refer to *Supplementary file 1B* for a summary of sample and population sizes, and for a statistical description (and comparisons) of each dataset in the magnetic coil system assays.

## Burrowing assays

We filled 5-ml plastic pipettes with 3% chemotaxis agar and cut and sealed the ends with Parafilm to minimize the formation of gaseous/humidity gradients similar to our previous study (*Beron et al., 2015*). We made three equidistant holes 10 cm apart in the pipettes. We injected 50 never-starved (day-1) adults into the center hole, and 1.5 μl 1-M of NaN$_3$ into the end holes to immobilize and easily count the worms that reached either side. Worms were first picked from their incubation plates into a 1-μl liquid NMG solution and transferred within 5 min into the center of the pipettes. Care was taken to ensure that worms were injected into solid agar (rather than remaining suspended in liquid solution once injected). The genotype of the strains was kept blind during the prep and running of the assay. The holes in the pipettes were then sealed with Parafilm. We wrapped the pipettes in aluminum foil multiple times (>3), to maintain the assays in complete darkness. Pipettes were aligned horizontally, in either the north-south or the east-west direction, or vertically in the up-down direction. Pipettes were placed inside a Faraday envelope made with copper cloth to prevent electric fields from interfering with the assays. The assays were allowed to run overnight and the worms immobilized at either end of the pipette were counted. The burrowing index (BI) was calculated as: BI = (A − B)/(A + B). Where A and B are the number of worms on opposite ends of the assay pipette.

## Burrowing in artificial magnetic fields

To assess if the surrounding magnetic field could disrupt burrowing behavior, we arranged burrowing pipettes vertically inside the magnetic coil system. We next generated a magnetic field of earth strength that had the opposite inclination (magnetic north up) to the local magnetic field (magnetic north down). Worms were allowed to burrow and their burrowing index was calculated as described above.

## Temperature gradient assessment

To determine if the presence of an artificial magnetic field in the Merrit Coil System produced a temperature gradient in the assay plate we used a Fisher High Accuracy thermometer sensitive to 0.01°C (*Figure 2—figure supplement 1*). We placed one probe on the center of the plate (where worms begin the assay) and one probe on the edge of the plate (where worms normally end the assay). After closing the Faraday cage, and with the coil system off, we measured the temperature on each probe in 5 min intervals for 20 min. At this point we powered the cage on to produce either a horizontal magnetic field of 0.650 Gauss (1× earth), or with a magnetic field able to cancel out the earth's own magnetic field inside the cage (field cancelled). We continued to record the temperature of the probes every 10 min for 1 hr (the duration of a typical coil system experiment). We next powered off the magnetic cage and once again recorded the temperature every 5 min for an additional 20 min (*Figure 2—figure supplement A,B*).

To determine if the magnets placed above the plates in our magnet assays produced a temperature gradient we placed temperature probes on the surface of the agar above the test and control positions of agar plates as indicated in the magnet assay procedure above. We compared the two temperature readings every 10 min for 1 hr *Figure 2—figure supplement C,D*). To determine if temperature gradients were created by the presence of a magnet we carried out these experiments both in the presence and in the absence of a magnet. Between temperature experiments, the thermal probes were immersed in a beaker containing 1 l of $dH_2O$ at the same distance they had during the experiments to determine if their reading differed from one another (*Figure 2—figure supplement E,F*).

## Testing genetically manipulated animals

Worms carrying extrachromosomal arrays with transgenes do not pass this construct to all their offspring. This permitted us to blindly test clones that are identical in their genetics and in their upbringing, only differing from one another in having (or not) the extrachromosomal array. In these experiments, a mixed population of worms (both carrying and not carrying the array) was tested together. After the assay, we used a fluorescent co-injection marker linked to the transgene of interest to identify and count the number of worms belonging to each population separately. We next calculated the magnetic orientation (or burrowing) index for each subpopulation of worms. This ensured that the comparisons were made between genetically identical populations that had grown under identical conditions, their only difference being whether or not they carried the extrachromosomal array. Expression of GCaMP3.0 did not interfere with ability to perform magnetotaxis to an artificial magnet (*Figure 7—figure supplement 1C*).

## Calcium imaging in restrained animals

Day-1 adult wild-type (and mutant) worms expressing the calcium-activity reporter GCaMP3 in neurons of interest were loaded with liquid NGM into a microfluidic chip (*Figure 7—figure supplement 1A*). Worms were immobilized and imaged in an Olympus BX51 scope at ×60 magnification. Images were sampled at 3.5–8 Hz using a CoolSNAP ES camera run by Windview 32. Each run lasted 50 s and begun with a 12.5 s of baseline followed by the presentation of a 6-s sinusoidal magnetic stimulus, and a 31.5-s recovery period. Consecutive recordings were made 3–5 min apart. A N42 Neodymium 3.5-cm diameter magnet (K&J Magnetics Inc., Pennsylvania, USA) was used to deliver the magnetic stimulus. The intensity of the stimulus was calibrated with a DC Milligauss Meter Model MGM magnetometer (AlphaLab, Utah, USA). The putative role of the magnetic sensor in the worm is to detect the direction of the magnetic field. It therefore follows that the cell must have an optimal stimulation angle between its sensor and the surrounding field. Because we could not infer what this optimal alignment angle may be, we decided to rotate the stimulus vector throughout 360° to ensure that each worm was stimulated with its presumably optimal angle. We did this by rotating the magnet along the xy plane followed by the xz plane at a rate of 2 Hz. A series of TIFF files was exported into ImageJ where the brightness of the cell body was measured across each photo series. The average soma brightness accounting for bleaching was calculated as previously described (*Kerr and Schafer, 2006*).

## Calcium imaging in partially restrained animals

Worms expressing GCaMP3 were incubated as described above and placed along with 3-µl liquid NGM on a 10% agar pad on a microscope slide with the coverslip pressed down to inhibit swimming, but permit slow crawling. Neurons were only imaged when the cells remained in the focal plane for the duration of the experiment. We placed the slide in an upright Olympus BX53 microscope equipped with and Retiga 2000R Fast 1394 camera (Q-Imaging, BC, Canada). Worms were illuminated with a Series 1200 UV light source (X-cite, Cincinnati, Ohio). We took a series of four pictures before, during, and after exposing the setup, for 10 s, to a 60 Gauss magnetic field generated by a Neodymium magnet. The Tiff files of the images were exported as 8-bit files. We used ImagePro 6.0 (MediaCybernetics, Rockville, MD) to measure the brightness of the soma in each picture series. We averaged the eight images before and after magnet exposure and compared this value to the average soma brightness of the four images taken during magnet exposure. We reported the percent change in brightness of the test condition to the average of before and after (*Figure 7—figure supplement 1B*).

## Image manipulations

All plots were graphed using SigmaPlot 12 (Aspire Software) and Matlab R2013b (Mathworks). Multiple plates were assembled in CorelDRAW X6 (Corel).

## Statistical analysis

All bars correspond to means, and variation is given as SEM throughout. Linear statistical analyses were performed using SigmaPlot 12 (Aspire Software). Comparisons between different experimental groups were performed by planned, two-tailed paired or unpaired $t$-tests to compare different groups that were normally distributed. Differences between non-normally distributed groups (or groups that failed the test of equal variance) were evaluated using the Mann–Whitney Ranked Sum Test, and two way repeated measures ANOVA (Temperature experiments). Correlations between parameters were determined using linear regressions and assessed using Pearson product–moment correlation coefficients. Circular statistical analyses (descriptive and comparative) were performed using a circular statistics toolbox for Matlab 2013b (*Berens, 2009*). We tested the significance of mean directions using Rayleigh tests, and for difference between vertical 'up' or 'down' direction and the mean direction of the population using V tests (*Batschelet, 1981*). Throughout this study, p values were reported using the convention: * $p < 0.05$, ** $p < 0.001$. Please refer to *Supplementary file 1A–E* for descriptive, and *Supplementary file 1A,B* for comparative statistics.

## Acknowledgements

We thank Drs C Bargmann, R Baumeister, I Mori, V Maricq, S Lockery, L Looger, and S Shaham for strains and constructs; G Thomas of the UT Austin Physics Electronic shop for assistance with magnetic coil system design; J Cohn for transgenesis assistance. Some strains were provided by the *Caenorhabditis* Genetic Center, which is funded by NIH Office of Research Infrastructure Programs (P40 OD010440) and the National Bioresource Project of Japan. Funding was provided by UT Austin UR Fellowship to KW and by a NIH NINDS grant to JP-S.

## Additional information

### Funding

| Funder | Grant reference | Author |
| --- | --- | --- |
| National Institutes of Health (NIH) | R01NS075541 | Jonathan Pierce-Shimomura |
| University of Texas | Undergraduate Research Fellowship | Kristi Ward |

The funders had no role in study design, data collection and interpretation, or the decision to submit the work for publication.

## Author contributions
AV-G, Contributed to experimental design, All experiments, and manuscript writing; KW, Performed magnet and magnetic coil system assays as well as coil system construction; CB, Contributed to burrowing experiments and magnet assays; NG, SG, Contributed to calcium imaging of immobilized worms; JR, Contributed to strain construction; NT, Contributed to burrowing assays and magnet assays; AP, Carried out burrowing assays; OG, Performed temperature experiments; AB-Y, Contributed to microfluidic chip design, and calcium imaging experiments; JP-S, Principal Investigator and contributed to experimental design; Writing of the manuscript; and unrestrained GCaMP3 experiments

## Author ORCIDs
Andrés Vidal-Gadea, http://orcid.org/0000-0001-5981-5528

## Additional files

### Supplementary file
• Supplementary file 1. (**A**) Results of statistical comparisons between groups tested in the magnet, burrowing, and chemotaxis assays. Comparisons appear in the order that they were introduced in the text. Two-tailed *t*-tests were performed between normally distributed groups that had equal variance to test the difference between the populations' means. The Pearson product–moment correlation coefficient was used to determine the correlation between variables. To test difference between means belonging to non-normally distributed samples we used the non-parametric Mann–Whitney ranked sum test. All statistic measures are from comparisons with the group shaded in grey immediately above. Index refers to the mean (magnetic or burrowing) index for each population (see 'Materials and methods'). (**B**). Summary of the magnetic coil system results for wild-type animals from three different locations around the world and internal physiological status. Statistical tests were performed using the Circular Statistic toolbox for Matlab (*Berens, 2009*). Comparisons between different parameters are shown within boxes. (**C**). Summary of local magnetic field properties and magnet assay results for wild-type worms from twelve different locations around the world (*Figure 5*). The exact isolation location for the LKC34 (Madagascar) strain is not known. We chose the northernmost location (Antisiranana) for the field properties, although choosing the southernmost location (Toliara) would not have yielded significantly different results. (**D**). Summary of the magnet-assay results for strains genetically modified to ablate (or to rescue gene function) in selected neurons (*Figure 6*). (**E**). Summary of the burrowing assay results (*Figures 1–3, 6*, *Figure 7*).

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
