## [Decision Letter]

Thank you for sending your work entitled “Magnetosensitive neurons mediate geomagnetic orientation in *Caenorhabditis elegans*” for consideration at *eLife*. Your article has been favorably evaluated by Eve Marder (Senior editor), a Reviewing editor, and three reviewers.

Summary:

In the field of magnetoreception in animals mechanisms suggested have been either magnetite or the radical pair mechanism of magnetoreception. However, neither the precise location of the magnetosensor nor its mechanism of action has been clearly elucidated in any animal to date. The data presented here convincingly show that *C. elegans* can use the Earth's magnetic field for its vertical “migrations” and that those migrations are influenced by satiation state; the movements are random when the field is cancelled and are clearly responsive to changes in inclination angle and intensity. These data provide rigorous, compelling data suggesting *C. elegans* as a new genetic system for studying magnetoreception.

Essential revisions:

The reviewers agree enthusiastically that this is an elegant demonstration of magnetoreception in *C. elegans* that has evolved to produce a clear magnetic sense. Their data show that the magnetosensor is housed in the AFD sensory neurons although the actual magnetosensing element is not yet identified. The persistence of magnetic orientation related to the origin of the species tested (e.g., Bristol) should be mentioned as suggesting genetic encoding of the magnetosensation.

Two negative controls would add to the paper:

For role of AFD: were there any experiments rescuing *tax-2/tax-4* in other neurons it is normally expressed in (other than AFD) or in neurons not normally expressing the channel and showing that they could not respond to magnetic fields? This relates to the necessity and sufficiency of AFD and/or *tax-2/tax-4*. If not, mention the possibility of other neurons playing a role.

For calcium imaging: in Figure 7—figure supplement 1 the authors have recorded from AWC and found it was not activated by magnetic fields. This is an important negative control and should be presented in a main figure as it shows that not all neurons respond to magnetic fields—an important point in the authors' arguments that AFD is the critical neuron for this behavior.

---

## [Author Response]

*The reviewers agree enthusiastically that this is an elegant demonstration of magnetoreception in* C. elegans *that has evolved to produce a clear magnetic sense. Their data show that the magnetosensor is housed in the AFD sensory neurons although the actual magnetosensing element is not yet identified. The persistence of magnetic orientation related to the origin of the species tested (e.g., Bristol) should be mentioned as suggesting genetic encoding of the magnetosensation*.

Thank you for your appreciation of our study. We have added the sentence “Our finding that the strains from England, Australia and Hawaii each displayed a preferred magnetic orientation preference that matched the geomagnetic field orientation at their source of isolation rather than the one at the experimental site (Texas, USA) suggests a genetic encoding of magnetic orientation preference” to the Discussion section.

*Two negative controls would add to the paper*:

*For role of AFD: were there any experiments rescuing* tax-2/tax-4 *in other neurons it is normally expressed in (other than AFD) or in neurons not normally expressing the channel and showing that they could not respond to magnetic fields? This relates to the necessity and sufficiency of AFD and/or* tax-2/tax-4*. If not, mention the possibility of other neurons playing a role*.

Thank you for this suggestion. To test this idea, we tested whether rescue of the *tax-4* gene in sensory neuron classes other than AFD rescued magnetic orientation behaviors. This strain was kindly provided by Dr. Ralf Baumeister. We found that although other *tax-4* dependent phenotypes were rescued (diacetyl chemotaxis), magnetic orientation behaviors were not rescued in this strain. Please find these results in Figure 6. This provides additional evidence for the AFD neurons playing a major role in magnetosensation.

*For calcium imaging: in*
Figure 7—figure supplement 1
*the authors have recorded from AWC and found it was not activated by magnetic fields. This is an important negative control and should be presented in a main figure as it shows that not all neurons respond to magnetic fields—an important point in the authors' arguments that AFD is the critical neuron for this behavior*.

We agree that this is an important control. We performed additional calcium imaging of the AWC neurons and added this data to Figure 7. AWC did not show a significant change when presented with magnetic stimuli consistent with the idea that the AFD neurons represent the primary magnetosensory neurons.